# Optical coherence tomography quantifies gradient refractive index and mechanical stiffness gradient across the human lens

Sabine Kling [1,2] ✉, Matteo Frigelli[2], M. Enes Aydemir[3], Vahoora Tahsini[2], Emilio A. Torres-Netto [3,4,5], Leonard Kollros[3] & Farhad Hafezi[3,4,5]

## Abstract

**Background** As a key element of ocular accommodation, the inherent mechanical stiffness gradient and the gradient refractive index (GRIN) of the crystalline lens determine its deformability and optical functionality. Quantifying the GRIN profile and deformation characteristics in the lens has the potential to improve the diagnosis and follow-up of lenticular disorders and guide refractive interventions in the future.

**Methods** Here, we present a type of optical coherence elastography able to examine the mechanical characteristics of the human crystalline lens and the GRIN distribution in vivo. The concept is demonstrated in a case series of 12 persons through lens displacement and strain measurements in an age-mixed group of human subjects in response to an external (ambient pressure modulation) and an intrinsic (micro-fluctuations of accommodation) mechanical deformation stimulus.

**Results** Here we show an excellent agreement between the high-resolution strain map retrieved during steady-state micro-fluctuations and earlier reports on lens stiffness in the cortex and nucleus suggesting a 2.0 to 2.3 times stiffer cortex than the nucleus in young lenses and a 1.0 to 7.0 times stiffer nucleus than the cortex in the old lenses.

**Conclusions** Optical coherence tomography is suitable to quantify the internal stiffness and refractive index distribution of the crystalline lens in vivo and thus might contribute to reveal its inner working mechanism. Our methodology provides new routes for ophthalmic pre-surgical examinations and basic research.

## Plain language summary

The lens of the eye changes in shape to enable objects at different distances from the eye to be seen clearly. Loss of ability to change the eyes' focus occurs during aging. We have developed a new way to image the eye that assesses how different lens regions change their shape. We evaluated our approach on twelve people of different ages and showed that those who were older had a stiffer lens, particularly in the central part of the lens. Further development and testing of our method could enable it to be used to both improve routine eye assessments as well as enable more research into how the eye works.

Accommodation is the process of the crystalline lens changing its shape and refractive power to adjust the eye's focus to a closer distance. This change is strongly dependent on the underlying optical and mechanical properties of the lens. The gradient refractive index (GRIN) of the crystalline lens has been extensively investigated in the past and there is evidence that it enhances the dioptric change during lens deformation[1] facilitating accommodation. Different indirect approaches have been adopted to measure the GRIN profile and its changes with age, either relying on the fact that the refractive index (RI) linearly correlates with protein concentration[2] in magnetic resonance imaging[3,4], by measuring the accumulated geometrical distortion induced by the GRIN in optical coherence tomography images[5–7], or by measuring the discrepancy in optical power measured experimentally versus the one expected from its geometry under the assumption of a homogenous RI distribution[1]. Yet, up until now, the GRIN has not been directly measured in the living eye.

The mechanical stiffness gradient of the crystalline lens has received much less attention than the GRIN. Mechanical properties determine the deformability of a material and thus play an essential role to which extent the lens can change its focal length, especially during the onset of presbyopia. Presbyopia is a progressive age-related insufficiency of optical accommodation culminating in the complete loss of the ability to focus on close distances above the age of 60 years. It originates most likely from a stiffening of the lens nucleus[8] (lenticular sclerosis[9]), yet up to date, only indirect evidence has been collected[10].

[1]Institute for Biomedical Engineering, ITET Department, ETH Zurich, Zurich, Switzerland. [2]ARTORG Center for Biomedical Engineering Research, University of Bern, Bern, Switzerland. [3]ELZA Institute AG, Dietikon, Switzerland. [4]CABMM, University of Zurich, Zurich, Switzerland. [5]Faculty of Medicine, University of Geneva, Geneva, Switzerland. ✉e-mail: kling.sabine@gmail.com

There exist different theories about the mechanism of accommodation: (i) Helmholtz's theory[11], which is most widely accepted, according to which the relaxation of zonules allows the lens to increase its curvature and the anterior lens to move forward, resulting in an increased thickness and decreased equatorial diameter. It was later refined by Fincham, suggesting that the lens capsule molds the lens into the accommodated shape and that the posterior lens moves backward. (ii) Coleman's theory requires a support force by the vitreous resulting from a positive differential pressure between the aqueous and the vitreous chambers. In fact, a pressure increase in the vitreous and a pressure drop in the aqueous in response to stimulated accommodation in primates has been described[12]. (iii) Schachar's theory[13], according to which the equatorial zonular traction causes the lens diameter to increase, the central curvature to steepen, the peripheral curvature to flatten and the central lens thickness to increase during accommodation.

So far, our understanding of accommodation and presbyopia is based on lens biometry and correspondingly on observations of lens thickness[14], curvature and ciliary body diameter[15] changes during stimulated accommodation. According to ocular biometry[16] during accommodation, the axial lens thickness increases by 320 μm in the emmetropic eye when changing from the far to the near point. The corresponding force of the ciliary muscle necessary to achieve this deformation has been experimentally determined to increase from approx. 9.1 to 12.8 mN between the ages of 15 and 45 years, and afterwards to decline to 10.3 mN at the age of 60 years[17]. In contrast, inverse simulation studies predicted a higher necessary ciliary force of 0.080 N[10,18]. At the same time, thickness changes due to micro-fluctuations of accommodation[19] have been reported to range between 20 and 35 μm and thus amount approx. 1/10th of the full accommodation range. A challenge in assessing internal lens deformations and displacements is the high transparency of the tissue resulting in a poor signal-to-noise ratio in structural imaging, including optical techniques, MRI[20] and ultrasound[21,22]. Therefore, most analyses are limited to analyzing anterior and posterior lenticular surface deformations, which inherently have the highest contrast. Previous literature has also aimed to identify the mechanical properties of different lens parts, mostly in combination with inverse numerical modeling[23,24]. Spinning tests performed on ex vivo human lenses of different ages suggested that the lens cortex is stiffer than the nucleus at all ages[25], and there is an age-related stiffness gradient[26].

More recently, Brillouin microscopy and micro air-puff-based OCT elastography were jointly applied to quantify the stiffness of ex vivo porcine lenses[27]. The wave propagation speed and Brillouin modulus strongly correlated when comparing the average stiffness derived with both techniques. However, it is surprising that Brillouin microscopy applied to characterize in vivo human crystalline lenses[28] suggests an opposite stiffness gradient than earlier predictions based on macroscopic lens deformation during accommodation and inverse numerical modeling[24].

In non-invasive corneal mechanical characterization, ambient pressure modulation[29,30] has been demonstrated as a suitable mechanical stimulus for optical coherence tomography (OCT) -based elastography under close-to-physiologic[31] stress conditions. While typically the cornea is static and maintains its shape extraordinary well under different environmental conditions, the lens is an inherently dynamic optical element, which constantly undergoes micro-fluctuations[32] of accommodation. The latter have been speculated to improve depth of focus[33] and to be involved in accommodation[34].

In the current study, we evaluate both, an external ambient pressure change and the naturally occurring steady-state micro-fluctuations of accommodation, as two distinct mechanical stimuli for their suitability to be used in the context of optical coherence elastography in the in vivo human crystalline lens. At the same time, we demonstrate the possibility of determining the GRIN from the same data set. We show that the assessment of micro-fluctuations allows a more refined analysis of the lens deformability compared to the ambient pressure-based analysis. The high-resolution strain maps retrieved during steady-state micro-fluctuations suggest a 2.0 to 2.3 times stiffer cortex than nucleus in young lenses and a 1.0 to 7.0 times stiffer nucleus than cortex in the old lenses. We confirm the GRIN to be lowest in the lens periphery (cortex) and highest in the center (nucleus).

## Methods

### Data set

Previously collected data sets[30,35] from our recent in vivo elastography studies in healthy participants (approved by the ethics committee of the ETH Zurich) and progressive keratoconus patients (approved by the cantonal ethics committee Zurich, number 2021-02275) were anonymized and re-examined with a focus on the crystalline lens. Keratoconus patients had received corneal cross-linking (CXL) treatment, but the crystalline lens is expected not being affected by this history. The rationale behind using this data set was its availability to the authors. The only demographic parameter that was preserved was the age category of the person (intervals of 5 years), and the only health-related information that was preserved is whether keratoconus has been diagnosed or not. Exclusion criteria were any pre-existing corneal disease other than keratoconus, anatomical constraints which prevent the correct placement of the measurement devices, pregnancy, a minimal corneal thickness of less than 350 μm or inability to understand the nature of the study and/or give consent. The study was conducted in accordance with the Declaration of Helsinki and informed consent was obtained. Informed consent was obtained for the OCT measurements described in the next paragraph to be undertaken from the people whose eyes were imaged, originally with the aim to study corneal changes associated with keratoconus. For extended data examination, the dataset was anonymized and retrospectively analyzed. The latter received ethical approval through a non-substantial amendment, with no need for additional patient consent. The whole data set was inspected and only those measurements retained in which at least 95% of the lens thickness were visible in the OCT images. This resulted in a total of twelve eligible measurements in twelve persons aged 20–25, 25–30, 30–35, 40–45, 50–55 and 70–75 years, with each age category being represented by two individuals, and a healthy: keratoconus ratio of 1:1.

Measurements were conducted with a spectral-domain commercial anterior segment optical coherence tomography (OCT) device (ANTE-RION, Heidelberg Engineering, Germany) with an axial and lateral resolution of 9.5 μm (in air) and 30 μm in the structural image, respectively, which was synchronized with an external pressure modulation unit as described earlier[30]. Briefly, each person was asked to wear a set of customized swimming goggles made of polycarbonate with a refractive index of 1.586 and a thickness of the front window of approx. 2 mm for the examination. The dispersion in the OCT images induced by the goggles was considered negligible. Then, 128 subsequent OCT B-scans were recorded with a frame rate of ~49 Hz (A-line rate of 50 kHz), while at approx. 1/3 of the recording time period, the pressure within the goggle chamber was suddenly reduced by 10 mmHg, inducing an expansion of the eye globe. During the whole examination period, the participant was looking at the internal fixation light (maltese cross) of the Anterion OCT system, which is designed to measure the eye during far vision and at zero vergence. As the person was focusing on the fixation light already during OCT positioning and alignment, we may assume that the accommodative state of the crystalline lens was stable when the measurement was taken. The dispersion corrected raw OCT data were exported and subjected to customized post-processing described in more detail below.

### Structural and elastographic interpretation

Figure 1 presents an overview of the different post-processing analyses conducted. Phase-sensitive axial displacement tracking was applied on the complex dispersion-corrected OCT signal following a vector-based summation approach described earlier in more detail[31,36]. Briefly, the optical pathlength change was computed from:

$$\Delta z = \frac{\lambda_0 \cdot \angle R}{4\pi \cdot n}, \tag{1}$$

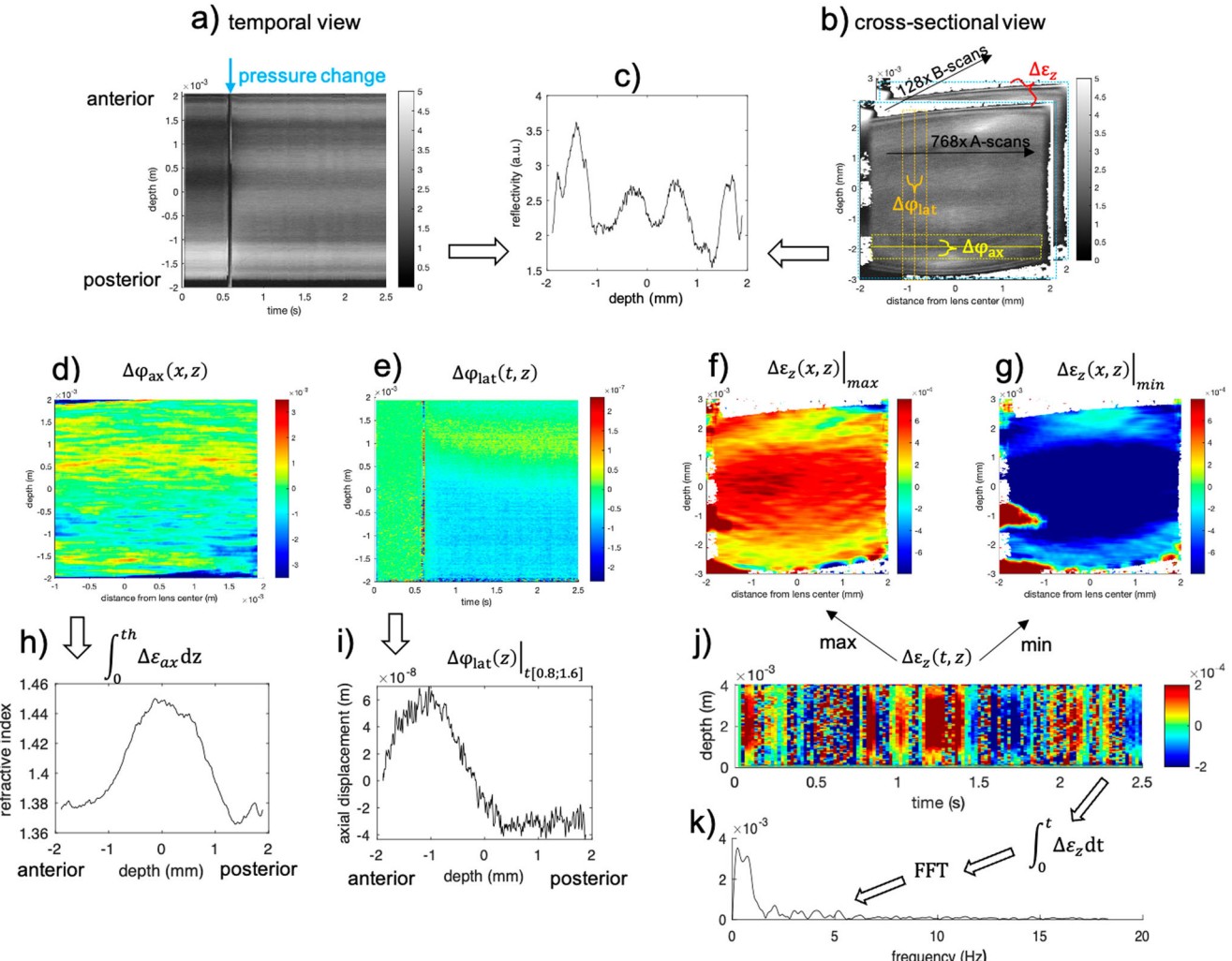

**Fig. 1 | Overview of the data post-processing pipeline, illustrated at a representative case. a** Temporal and **b** cross-sectional views of the motion-correction structural raw OCT data. **c** Structural reflectivity profile. **d** Axial, **e** lateral and **f, g** cross-sectional phase difference tracking. **j** Temporal strain fluctuation. Mathematical operations and correspondingly derived parameters: **h** refractive index distribution, **i** axial displacement, **k** frequency assessment.

where $\lambda = 1300$ nm is the mean wavelength of the OCT, $n = 1.41$ the equivalent, homogeneous refractive index of the human lens[5], and $\angle R$ the angle of the complex cross-correlation R.

**Refractive index mapping.** The axial phase difference between two neighboring axial pixels within a single A-scan represents the change in optical path length ($OPL = \sum n_i L_i$) due to the shift by one pixel, according to:

$$\Delta\varphi_{ax} = \frac{4\pi}{\lambda_0} n \cdot L, \qquad (2)$$

By expressing the refractive index $n = n_c + \Delta n_{ax}$ in dependency of a reference region with known refractive index $n_c$ (e.g. the cornea with $n_c = 1.375$), the axial refractive index distribution is obtained via:

$$\Delta n_{ax} = \frac{\lambda_0 \cdot \Delta\varphi_{ax}}{4\pi \cdot L_0} - n_c = \frac{\lambda_0 \cdot n_c}{4\pi \cdot L_0}\left(\Delta\varphi_{ax} - \Delta\varphi_c\right), \qquad (3)$$

Where $\Delta\varphi_c$ corresponds to the axial phase difference measured in a reference region with known refractive index $n_c$ and $L_0 = L_c \cdot \frac{n_c}{n_0}$ is the pixel size in air, with $n_0 = 1$. Subsequently, the axial refractive index is retrieved by

integrating the axial phase shift according to:

$$n_{ax} = \int dn_{ax}(z)dz + n_c, \qquad (4)$$

In order to account for noise in the measurement signal, we made the assumption that the gradient refractive index is symmetrically distributed, such that the sum of the axial gradient of the refractive index within the lens equals zero across an A-line. This is a reasonable assumption since the refractive indices at the anterior and posterior lens surfaces have the same value[37].

**Displacement tracking.** The lateral phase difference $\Delta\varphi_{lat}$ represents the change in optical path length due to both, a minimal change in the refractive index $\Delta n_{lat}$ between neighboring pixels and a minimal change in the OPL resulting from displacement or geometrical differences:

$$\Delta\varphi_{lat} = \frac{4\pi}{\lambda_0}\left[L_0 n_0 + L_c n_c + L_a n_a + L_x n_x - (L_0 - \Delta L_c)n_0\right.$$
$$- (L_c + \Delta L_c - \Delta L_a)n_c - (L_a + \Delta L_a - \Delta L_x)n_a$$
$$\left. - (n_x - \Delta n_{lat})(L_x + \Delta L_x)\right], \qquad (5)$$

**Table 1 | Material properties of the different ocular tissues assigned for the finite element simulation**

|  | C1(32.5y) | C3(32.5y) | C4(32.5y) | D22.5 |
|---|---|---|---|---|
| Lens capsule | 216 kPa | 33.9 kPa | 9.5 kPa | $0.284 \times 10^{-6}$ |
| Lens cortex | 451 Pa | - | - | $34.5 \times 10^{-6}$ |
| Lens nucleus | 226 Pa | - | - | $215 \times 10^{-6}$ |
|  | C1 | C2 | C3 | D |
| Cornea | 35.5 kPa | 3.2 kPa | 1.9 kPa | $10^{-8}$ |
| Sclera | 810 kPa | 56 MPa | 2332 MPa | $10^{-8}$ |

For the lens, C1 is the Neo-Hookean material constant and C3, C4 represent the the stiffness of the preferential direction of deformation. For cornea and sclera, C1, C2, C3 are the Yeoh material constants. D is the incompressibility parameter.

Where subindex $0$ refers to air, $c$ to cornea, $a$ to anterior chamber and $x$ to the lens. $L$ indicates the physical length and $n$ the refractive index of the given structure. By subtracting the mean lateral phase difference across the full lens thickness and under the approximation that $\Delta n_{\text{lat}} \Delta L_x \to 0$, Eq. 5 can be reduced to:

$$\Delta\varphi_{\text{lat}} = \frac{4\pi}{\lambda_0} \left[ n_x \Delta L_x + L_x \Delta n_{\text{lat}} \right], \tag{6}$$

As the two terms $n_x \Delta L_x$ and $L_x \Delta n_{\text{lat}}$ are expected to contribute to a similar extent to the measured lateral phase difference, a symmetry assumption on the gradient refractive index distribution was made, namely that the sum of the lateral gradient of the refractive index equals zero within the central lens region. Thus, the average axial displacement during an entire B-scan results from the sum of the lateral phase changes in lateral direction ($\overline{\Delta L_x}$):

$$\Delta\bar{L}_x = \sum_x \frac{\Delta\varphi_{\text{lat}} \cdot \lambda_0}{4\pi \cdot n_x}, \tag{7}$$

The lateral phase difference $\Delta\varphi_{\text{lat}} = \angle R_\alpha$ was determined by computing the complex cross-correlation of two subsequent A-scans ($R_\alpha$), recorded with a pixel overlap of $\sim 65\%$, according to:

$$R_{\alpha,S}(z, x) = \sum_{j=-u_z}^{u_z} \sum_{k=-u_x}^{u_x} B_S(z+j, x+k) \cdot B_S^*(z+j, x+k+1), \tag{8}$$

where B represents an OCT B-scan, B* indicates its complex conjugate, and s = {1, 128} is the number of B-scans, z = {1, 1222} is the number of axial pixels and x = {1, 768} the number of lateral pixels. We used phase-processing windows of the size $u_z = 2$ and $u_x = 5$, resulting in a resolution of 28 μm x 104 μm for displacement tracking. Before computing the cross-correlation, the anterior surface of the lens was detected and each B-scan was converted to show a flat sample. Subsequently the rigid body displacement of the lens (average across whole lens thickness) was subtracted and the signal was laterally averaged across the 1 mm central optical zone. Correspondingly, the retrieved axial displacement profile presents the relative lens deformation $\overline{\Delta L_x}$.

**Strain tracking.** In order to take advantage of the high spatial resolution of the OCT images and derive strain maps $\Delta\varepsilon_z(z, x)$ across the cross-section of the whole crystalline lens, complex cross-correlation was computed between two subsequent B-scans ($R_\beta$) taken at the same location, separated by the time $\Delta t \sim 21$ ms:

$$R_{\beta,S}(z, x) = \sum_{j=-w_z}^{w_z} \sum_{k=-w_x}^{w_x} B_S(z+j, x+k) \cdot B_{S+1}^*(z+j, x+k), \tag{9}$$

where B represents an OCT B-scan, B* indicates its complex conjugate, and s = {1, 128} is the number of B-scans, z = {1, 1222} is the number of axial pixels and x = {1, 768} the number of lateral pixels. We used phase-processing windows of the size $w_z = 7$ and $w_x = 7$, resulting in a resolution of 97 μm x 146 μm in the cross-sectional displacement maps. Before computing the cross-correlation, subsequent B-scans were re-aligned by 2D cross-correlation in order to remove motion artefacts. Subsequently, axial strain as represented by the axial gradient of Δz, was computed by a second cross-correlation according to:

$$\Delta\varepsilon_z(z, x) = \frac{\lambda \cdot \angle \sum_{l=-v_z}^{v_z} \sum_{m=-v_x}^{v_x} (R_{\beta s}(z+l, x+m) \cdot R_{\beta s}^*(z+1+l, x+m))}{4\pi \cdot \delta z}, \tag{10}$$

where $\delta z = 9.5$ μm defines the axial sampling unit (in air). $v_z = 15$ and $v_x = 15$ were the applied strain processing windows, resulting in a resolution of 207 μm x 313 μm in the cross-sectional strain maps. As can be seen from (10), axial strain measured with OCE is independent of the refractive index, which is in strong contrast to the structural OCT image and the derived displacement Δz. Due to this particular characteristic, OCT elastography measurements are not biased by the inherent gradient refractive index of the crystalline lens. In order to assess the temporal dynamics of the micro-fluctuations, the instantaneous strain within the lens region of a single B-scan was averaged and subsequently integrated in time, before the Fourier transform was applied to determine the dominant oscillation frequency.

**Finite element modeling**
In order to better understand the induced lenticular deformations measured in response to both, external under-pressure stimulation ($\overline{\Delta L_x}$) and ciliary muscle actuation ($\Delta\varepsilon_z(z, x)$), an axisymmetric finite element model was built in ANSYS software (Mechanical APDL, Build 22.1, ANSYS Inc.) to reproduce the experimental setting. A similar lens geometry as in previous literature[10] was applied and meshed with PLANE183 elements. The outer ocular wall was modelled with SHELL208 elements. The aqueous and vitreous humors were meshed with FLUID29 elements and subjected to a normal intraocular pressure of 15 mmHg. A Neo-Hookean material model was applied for the lens and a Yeoh material model for the cornea[38] and sclera[39]. Material constants according to Table 1 were assigned. To determine the material constants of the lens, first the same material parameters as retrieved in a recent inverse simulation study were applied for the 32.5-year-old lens. We verified that with this setting the same accommodation amplitude of 7.5 diopters as in the original publication[10] was achieved, which is representative for this age-group. Next, the ratio of the C1 parameter between nucleus and cortex was adjusted to match with the strain ratio found in the current study. Subsequently, an overall weaking factor of 0.75 (for the C1 parameter of nucleus, cortex and capsule) was introduced to regain the same accommodative amplitude as before. In order to account for ageing in the different groups, a tissue stiffening factor derived from the ratio of the strain amplitude was experimentally retrieved from the current assessment of micro-fluctuations (section 3.3). This resulted in a factor of 0.81, 0.89, 1.00, 1.10, 3.73 and 15.3 for the nucleus and a factor of 0.85, 0.92, 1.00, 1.21, 1.78 and 3.37 for the cortex at the medium age of 22.5, 27.5, 32.5, 42.5, 52.5 and 72.5 years, respectively. In order to simulate micro-fluctuations of accommodation, the ciliary force was reduced until the induced strain amplitude was similar to the experimentally recorded one in the 32.5-year-old group. We approximated this ciliary force during micro-fluctuations to remain constant across age. To simulate the ambient pressure decrease, a negative pressure of −10 mmHg was applied on the anterior external ocular shell. To constrain the model, nodes located at the equator of the lens were restricted to move along the optical axis of the eye.

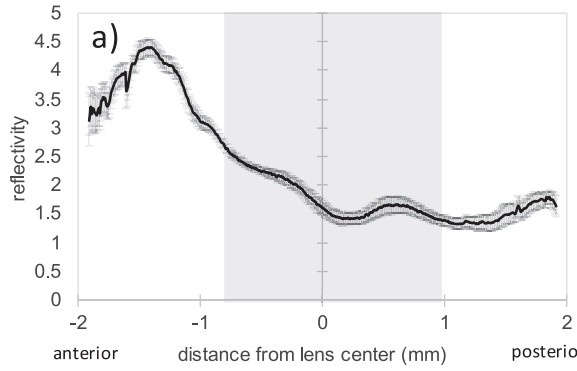

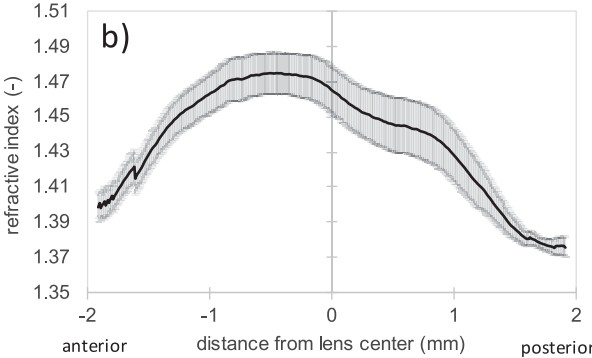

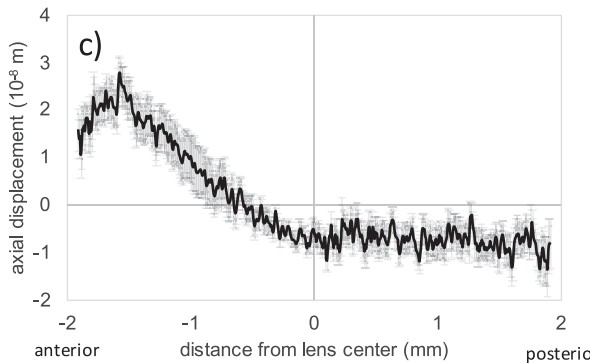

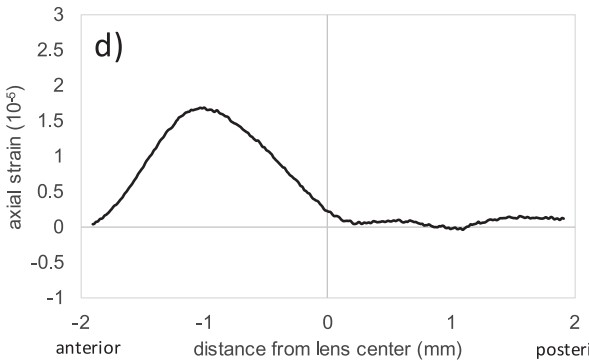

**Fig. 2 | Refractive and mechanical profiles averaged across the optical zone.**
**a** Structural reflectivity profile of the human crystalline lens as a function of depth.
The gray shaded area indicates the region, which was considered as the nucleus at all
ages. **b** Corresponding gradient refractive index distribution. **c** Corresponding dis-
placement and **d** the smoothed strain profile corresponding to the deformation in
response to an ambient pressure change by −10 mmHg recorded during ~ 0.6 s. In
panels **a**–**c**, error bars represent standard error across the full data set (n = 12 bio-
logically independent samples).

## Statistics and reproducibility

Statistical analyses were conducted in SPSS Statistics (IBM Corporation,
Version 28.0.1.1(14)). Sample size was $n = 12$ independent biological sam-
ples. Normality of the data distribution was assessed with the Shapiro-Wilk
test. Accordingly, Spearman correlation was applied to assess the relation-
ship between different parameters with a skewed distribution. The achieved
statistical power of the correlation analysis was 85%. To compare the
difference in the refractive index between anterior and posterior regions
of the lens (skewed distribution), a two-tailed related samples Wilcoxon
signed rank test was applied. $P$-values of <0.050 were considered to indicate
statistical significance. The achieved statistical power of this analysis
was 100%.

## Reporting summary

Further information on research design is available in the Nature Portfolio
Reporting Summary linked to this article.

## Results

For a better interpretation of the data, the lens was positioned with its
geometrical center at the origin of the coordinate system. Next, the geometry
was divided into cortex and nucleus based on the reflectivity profile of the
structural OCT image. The gray shaded area in Fig. 2a indicates the nucleus.
The remaining peripheral area was assigned as cortex.

### Gradient refractive index distribution

The refractive index profile within the crystalline lens could be well recov-
ered from the in vivo measurements, see Fig. 2b. On average, the refractive
index ranged between 1.376 and 1.475. Interestingly, the anterior nucleus
had a significantly ($p = 3.7 \times 10^{-5}$) higher refractive index than the posterior
nucleus, on average by $0.028 \pm 0.010$ (mean ± standard error) across the
whole data set with a medium effect size (Cohen's d = 0.68).

### Ambient pressure reduction

Reducing the ambient pressure in front of the eye results in a decrease of the
intraocular pressure (IOP) according to earlier literature[40], which in the
current study induced a characteristic deformation of the lens, both
experimentally and numerically. Experimental deformations were assessed
during approx. 0.6 s after the ambient pressure change. This time period was
chosen as blinking was absent in all participants.

When positioning the lens with its center at the origin of the coordinate
system, a similar displacement behavior was observed in all lenses (Fig. 2c),
with a maximal amplitude at −1.6 mm distance from the lens center
(towards the anterior surface). Notice that the net displacement was zero
because rigid body motion was subtracted. The current study did not
investigate the relative lens displacement with respect to the cornea. No
significant correlation was observed between age and the maximal axial
displacement ($p = 0.544$). There was also no detectable change in the
anterior lens curvature in response to the pressure stimulus.

The gradient of the axial displacement (Fig. 2d), which was computed
after Gaussian smoothing with a window size of 200 pixels, corresponds
to the induced axial strain. The overall positive strain in response to ambient
pressure decrease suggests an axial expansion (relaxation), which is
in agreement with accommodation in response to the under-pressure sti-
mulus. There was a trend toward lower axial deformations and strain
with age.

### Micro-fluctuations of accommodation

Throughout the repeated B-scan acquisition duration of ~ 2.0 s, physiologic
steady state micro-fluctuations of accommodation independent of the
external pressure application were observed. In this context, micro-
accommodation was defined as the lens becoming thicker and micro-
disaccommodation as the lens becoming thinner. It is worth mentioning
that lens micro-fluctuations were observed even at no accommodative
demand. Figure 3 shows representative snapshots of the strain profiles at
maximal amplitude during micro-accommodation and micro-dis-

**Fig. 3 | Cross-sectional structural and mechanical images. a**, **c**, **e**, **g**, **i** Representative cross-sectional views of the crystalline lens in structural and elastographic OCT images. **b**, **d**, **f**, **h**, **j** Elastographic images show the strain distribution at maximal strain amplitude during micro-accommodation (left half) and micro-dis-accommodation (right half). From top to bottom in each image there is the anterior lens cortex, lens nucleus and posterior lens cortex. In lenses ≤45 years the lens nucleus deformed most (presented largest strains) demonstrating a softer behavior compared to the cortex.

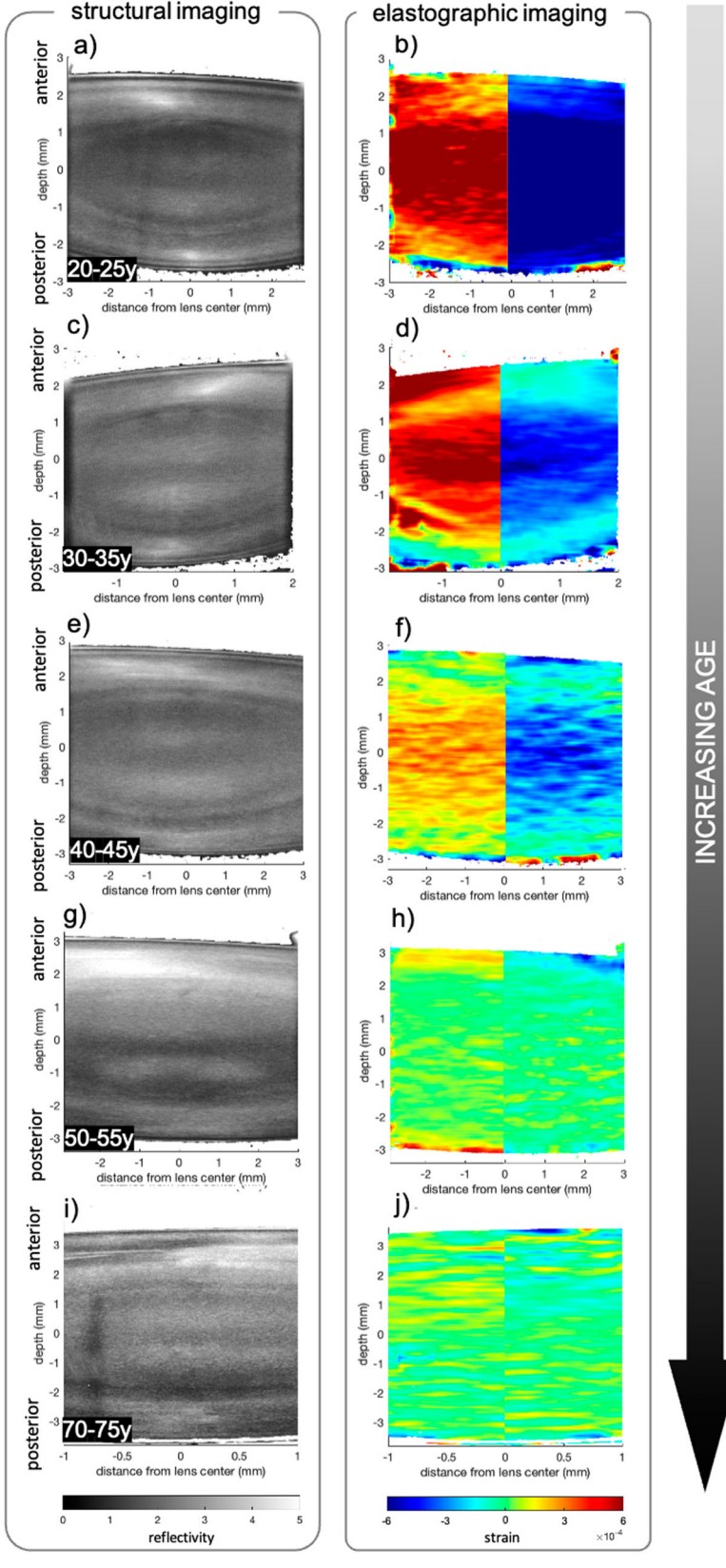

accommodation for different age groups. As can be recognized, in <45-year-old participants, the nucleus deformed strongest and the lens cortex least. The situation changes in the older participants, where the deformation was more concentrated on the lens cortex and the nuclear strain amplitude decreased dramatically.

Figure 4a–f shows a comparison of the axial strain profiles at different ages. The average strain across the crystalline lens during accommodation and dis-accommodation was strongly correlated with age ($r = 0.828$, $p = 0.00088$), see also Fig. 4g. On average, there was an axial strain between $14.2 \times 10^{-6}$ and $384 \times 10^{-6}$ across the full lens thickness during

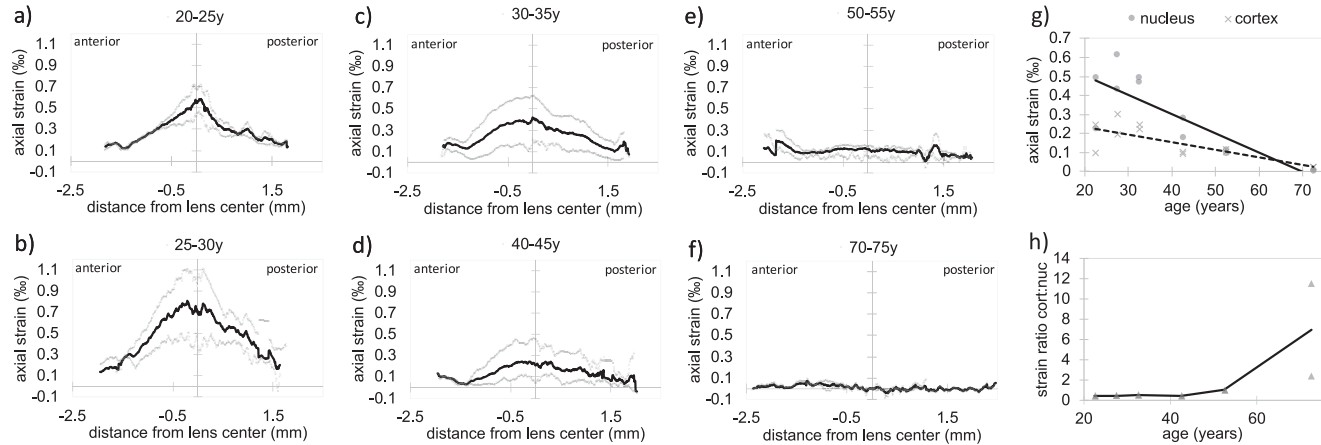

**Fig. 4 | Strain assessment from micro-fluctuations. a–f** Average axial strain profiles of the human crystalline lens during a micro-fluctuation obtained with OCT elastography. With increasing age, the strain amplitude in the nucleus decreases drastically and strains in the cortical region slightly increase. Gray dots represent the mean strain profile of an individual participant ($n = 2$ independent samples per panel). The black line shows the average strain profile for the respective age range. **g** Correlation between age range and the axial strain ($n = 12$ independent samples) induced in the nucleus and cortex, respectively. Gray markers represent single data points, the black continuous and dashed lines represent the average trend lines, respectively in the nucleus and cortex. **h** Strain ratio of cortex: nucleus as a function of age ($n = 12$ independent samples). A substantial change is observed between the age of 45 and 55, which corresponds to the onset of presbyopia. Gray markers represent single data points, the black line represents the average curve.

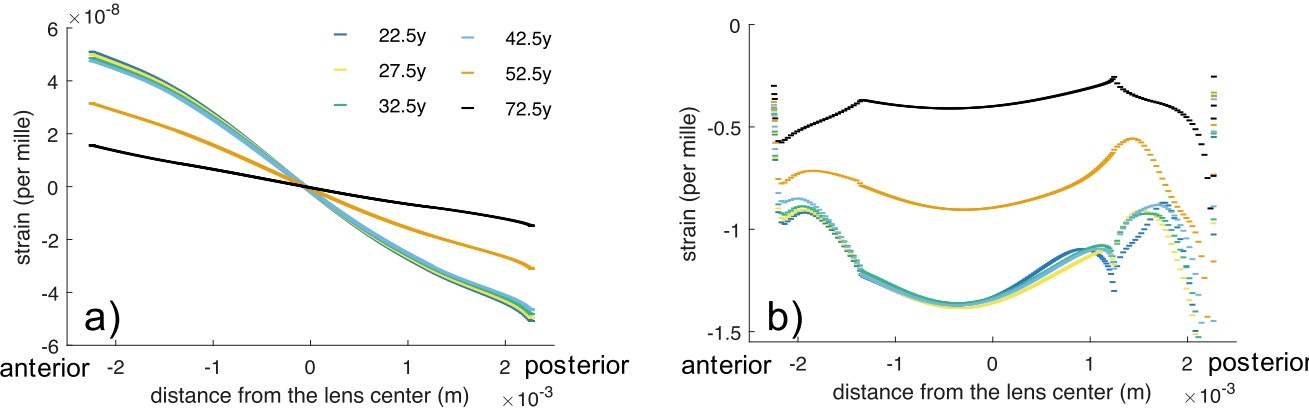

**Fig. 5 | Simulation results. a** Axial displacement profile in response to an ambient pressure decrease of 10 mmHg. For comparison with the experimental data, see Fig. 2c. **b** Axial strain profile during a micro-fluctuation with a ciliary force of 39 μN. For comparison with the experimental data, see Fig. 4a–f.

accommodation and between $19.5 \times 10^{-6}$ and $501 \times 10^{-6}$ during disaccommodation. This corresponds to a thickness change between 71.0 nm (old lens) and 2.00 μm (young lens) in the time between two subsequent B-scans ($\sim 20$ ms). The stiffness ratio $S_{cn}$ of the cortex: nucleus was defined under the assumption of a homogenous stress distribution within the lens:

$$S_{cn} = \frac{E_n}{E_c} = \frac{\Delta\varepsilon_c}{\Delta\varepsilon_n}, \qquad (11)$$

where $E_c$ and $E_n$ are respectively the cortical and nuclear stiffness, $\Delta\varepsilon_n$ is the average experimental strain in the nucleus, and $\Delta\varepsilon_c$ the average experimental strain in the cortex in a given age group. The stiffness ratio (Fig. 4h) was lowest in lenses of ≤45 years of age. The quantification of the micro-fluctuation frequency was constrained to an approximate value of 0.3 Hz, given that multiple B-scans within each measurement were compromised by ocular motion, resulting in the degradation of the phase reference, see Fig. 1j.

There was a correlation of the max: min structural reflectivity ratio with age ($c_{sp} = -0.60$, $p = 0.039$), however no relationship between reflectivity and strain.

## Numerical simulation

In terms of simulating an ambient pressure reduction, the induced axial deformation pattern showed a forward displacement in the anterior lens, and a backward displacement in the posterior lens (Fig. 5a), similarly to the experimental data. The induced average axial displacement amounted on average $\pm 4.1 \times 10^{-8}$ m, which was 2.3 times higher than the experimental value. The age-dependency (material stiffening) seemed to be more pronounced in the simulations than experimentally. Interestingly, the axial displacements (in the direction of the eye's optical axis) induced in the cornea (209 μm) and sclera (82 μm) were substantially higher compared to the deformations induced in the lens.

In terms of simulating micro-fluctuations, a reduction by a factor of 2000 in the maximal ciliary force (full accommodation) was necessary to achieve a similar nuclear axial strain amplitude (for the 32.5-year-old lens) as in the micro-fluctuation experiments. The simulated strain amplitude amounted 0.30‰ in the 22.5-year-old lens and declined to 0.07 ‰ in the 72.5-year-old lens, see Fig. 5b. Notice that in the experiment, the full deformation happened across ~3 subsequent B-scans, but only the scan with the max deformation is presented. Under the assumption of a sinusoidal deformation pattern, the full experimental deformation is 2x as much as Figs. 3 and 4 report. In contrast, the simulation results directly report the

full deformation. Overall, the strain amplitude demonstrated a similar decline with age as the experimental values, yet suggested the onset of the pronounced strain decrease at a slightly older age than the experiments.

## Discussion

We report a way of close-to-natural optical coherence elastography suitable to quantify internal lens displacements and strain in vivo and with high spatial resolution. Furthermore, we show that from the same data set and signal, the gradient refractive index distribution can be simultaneously retrieved. For the first time to our knowledge, we present the experimentally measured internal axial displacement profile arising from a change in the ocular pressure distribution along with the GRIN profile, and the internal axial strain pattern during physiological micro-fluctuations of accommodation, both in vivo in human subjects.

So far, the lens diameter is considered a valuable indirect measure whether the lens gets stretched or relaxes. Yet, the optical in vivo examination of lens accommodation is hampered by the presence of the iris preventing to image the peripheral lens. Only few cases of aniridia or albino[14] patients have been examined providing more insight on the peripheral lens deformations and the action mechanisms of the ciliary muscle during in vivo accommodation. The current study has a similar limitation in this regard given that only the deformation of the central crystalline lens could be assessed. However, with OCT elastography macroscopic changes in lens shape play a secondary role, given that internal tissue deformations can be directly accessed. The distinction between cortical and nuclear regions in the current study was made based on the reflectivity profile and the assumption that the dimensions of the nuclear region are comparable across ages, which forms another limitation. In the future, a more refined distinction of the cortical and nuclear regions might become possible based on the herein derived strain maps. Moreover, our phase-sensitive OCE approach provides an extraordinarily high sensitivity to quantify displacement and strain, yet the assessment is limited to the axial direction, which is another limitation of the present study. One factor to be considered in that context is that due to light refraction at the corneal and lens surfaces, the OCT beam interrogates the lens not orthogonally but under an angle of up to ±9° in the more peripheral lens close to the pupil. In consequence, the registered displacement contains up to 1/6 of the lateral lens displacement, which is twice as large as the axial displacement when assuming a Poisson's ratio[41] of 0.5. In essence, the presented values for axial deformation and strain are a slight overestimate. As the front surface of the goggles was flat, no optical distortion is expected from this additional layer. A further limitation is that the entire posterior lens surface was not always within the imaging range preventing a macroscopic validation of lens thickness changes. A limitation in this context is that accommodation has not been actively controlled or verified. Even though a visual target was presented for fixation during the measurement, we cannot guarantee that the participant had the target in focus at all time. Phase-sensitive detection itself poses a limitation with in vivo applications due to the high susceptibility to motion resulting in signal loss. Due to this effect, several frames in each acquisition were lost as shown in the visualization of $\Delta\varepsilon_z(t, z)$ in Fig. 1j. Although a better controlled experimental setting is desired for the future, to our knowledge the current study for the first time demonstrates the possibility of simultaneously recording the gradient refractive index and deformation behavior of the inner human lens in vivo. The retrospective inspection of available elastography measurements conducted in healthy participants and keratoconus patients allowed us to conduct the presented study without the need of acquiring new participants and conducting the same measurements. In future studies, a larger sample size and a sub-grouping accounting for ophthalmic diagnoses might be desirable.

The observed GRIN distribution, which ranged between 1.337 and 1.414 [95% CI] in the cortex and between 1.436 and 1.514 [95% CI] in the nucleus according to the current study, was higher compared to prior literature reporting ranges for the human crystalline lens between 1.378 and 1.410[6], 1.36 and 1.43[3] or 1.34 and 1.40[42]. The current values were closer to previously reported GRIN ranges for rabbit lenses with values between 1.431

and 1.464[43] or between 1.3656 and 1.504[44]. This discrepancy might be related to the long wavelength (1200 to 1400 nm) that was used for the measurement in the present study. However, the overall shape of the GRIN – with highest values in the lens nucleus and lowest values in the cortex – agreed well with earlier literature[6,23,42]. With a relatively low sample size and rather high levels of noise in the refractive index measurement, the current study did not show significant differences in the GRIN with age.

Our findings agree with earlier studies that suggest a softer nucleus than cortex in the young lens and the inverse stiffness distribution in the older lenses[8] above 50 years. While an absolute quantification of lens stiffness was not possible in the current study as the zonular forces are inaccessible in vivo, the fact that the ratio between cortical and nuclear strains changed with age confirms the lenticular theory[45] of presbyopia, according to which lens sclerosis or more general mechanical changes result in the loss of accommodation[15]. Would instead the lens properties remain unaltered and only the extraocular loading decrease in amplitude (extra-lenticular theory), the ratio between cortical and nuclear strains would have been stable across ages. Assuming a homogenous stress distribution along the optical axis, which is a reasonable approximation for the central lens according to the finite element model (Supplementary Fig. 1), we found a 2.0 to 2.3 times stiffer cortex than the nucleus in <45-year-old lenses and a 1.0 to 7.0 times stiffer nucleus than the cortex of lenses aged above. These values are in agreement with the lower limit of stiffness ratios reported by Weeber et al. [23], who observed a 2.2 to 16 times stiffer cortex than nucleus in young lenses, and a 5.8 to 210 times stiffer nucleus than cortex in old lenses. However, there are also different prior findings regarding the stiffness of the lens cortex with respect to the nucleus. For example, Hollman et al.[46] reported a 4.7 times stiffer nucleus than the cortex in middle-aged lenses. It should be considered that the vast majority of stiffness measurements of the human crystalline lens have been performed on post-mortem tissue, which might explain a higher variability in the observations and potentially systematic error arising from the chosen preservation method in lenses that were not immediately measured. Likewise, the degree of lens stiffness might be quite different from patient to patient. In other words, during a surgical lens fragmentation and removal through a cataract surgery, it is readily noticeable that patients of similar ages can have different degrees of cataracts. Therefore, the variation found could also be attributed to individual factors.

On a more absolute perspective, our results are in agreement with the observation that cortical stiffness continuously increases[25] until the age of approx. 50 years, while the nuclear stiffness remains relatively constant during this period. With increasing age, we observed a stiffening by a factor of 4 in the cortex and of 19 in the nucleus. An earlier study exploring the mechanical gradient in the aging human lens reported an age-related stiffening[8] of 20 in the cortex and of 450 in the nucleus. While the magnitude of the changes was smaller in the current study, both studies agree in the fact that the nucleus experiences a much stronger stiffening with age than the cortex. This is also in line with the authors' subjective perception when performing cataract surgery on older patients within the same geographical region.

The frequency of the observed micro-fluctuations (~0.3 Hz) comes close to the low-frequency components (< 0.6 Hz) of steady-state mechanical fluctuations of accommodation reported earlier from ocular biometry[47] using optical coherence tomography. Micro-fluctuations have earlier been quantified to amount[19] approx. ± 0.5 diopters with a corresponding thickness change of ± 20 to 35 μm. Considering the imaging frame rate of ~49 Hz and assuming a sinusoidal fluctuation, the corresponding thickness changes we may expect from our measurements is 38μm and thus similar.

The external pressure-decrease induced internal lens deformations that suggest axial lens relaxation, which – assuming that the IOP naturally exerts a compressive force on the lens – is in good agreement with the expected reduction of the IOP. The numerical simulation also demonstrated that the distance between lens and posterior sclera (i.e. retina) gets shortened by the applied pressure change, which furthermore induces the need of an

accommodative effort to keep the visual target in focus. The absence of this corrective accommodation stimulus in the numerical model might explain minor discrepancies between experiment and simulation. For example, in contrast to the numerical simulations, the experimentally-observed pressure-induced lenticular deformations seemed to be only weakly dependent on the lens' mechanical properties and did not reach statistical significance. The reason for this might also be found in a relatively high inter-person variability, which could arise from the fact that different tissue groups (lens, cornea, sclera) are involved in the pressure-induced deformations. Furthermore, due to the overall small displacement amplitude, the light-source related noise level in the pressure-related measurements could have prevented the detection of an age-related decline in the induced deformation. In contrast, our observations with respect to the steady-state micro-fluctuations were the most unexpected ones, but those with the highest relevance, as they demonstrate both, a significant correlation with age and a substantially lower inter-subject variability. As such, assessing the deformations induced during micro-fluctuations could present a straight forward approach to quantify the relative stiffness of the lens and this way test its biological age. In addition, compared to the pressure modulation approach, assessing micro-fluctuations has the advantage that no external mechanical stimulation is necessary. Assuming a relatively constant ciliary muscle force across age, which is a similar approximation as made in earlier simulation studies[10], the obtained strain maps as shown in Fig. 3, can be also be used for absolute material parameter identification via inverse numerical modeling. In the current study, we assumed the same factor of material stiffening, as the ratio between cortical and nuclear strains increased with age. This appeared to be a reasonable approximation, since the simulation results did reasonably well agree with the experimental data. However, a more sophisticated inverse modeling architecture is recommended for a more accurate material parameter identification in the future. Overall, the present study cannot confirm or reject any theory of accommodation mentioned in the introduction, as only the lens deformations, but not the acting forces have been measured.

So far, structural assessment of the crystalline lens has been limited to optical[16], ultrasound[48] and MRI[49] contrast. In this very small study population, we did not observe any correlation between the structural reflectivity profile and the mechanical deformation amplitude, thus indicating the elastographic examination provides distinct and previously inaccessible information. Due to the simplicity of assessing the GRIN and deformations due to micro-fluctuations already from a set of 128 repeated B-scans, the proposed methodology could be easily integrated into existing OCT devices. This functional imaging approach could also complement optical techniques assessing the accommodation amplitude of the lens with a spatially-resolved localization of age-related mechanical changes and thus inspire the design of accommodative artificial lenses in the future.

Brillouin microscopy is possibly the only previously applied technique that allowed an in vivo examination of the lens stiffness with high resolution[28]. However, the Brillouin observations suggest a stiffer nucleus than cortex at all ages and only the stiffer nuclear region to increase with age. This is in contrast to the findings of our present study and most previous literature[23,24,41,46], and might indicate a biased Brillouin measurement. The Brillouin signal depends on the refractive index of the material, but the GRIN has not been specifically addressed in these measurements[27,50]. In contrast, the OCT elastography measurements are not inherently biased by the refractive index and as such could permit a more realistic assessment of the lens deformability and thus its mechanics. The apparent difference between Brillouin and OCE measurements might be also be related to the viscoelastic material properties[23,51,52] of the lens. The stiffness of viscoelastic materials depends on the deformation speed and increases with higher frequencies. Recently, we could confirm the presence of viscoelasticity in the nucleus and quantify it in ex vivo porcine lenses[53]. Given that the Brillouin shift is measured at GHz frequencies, the stiffness at lower (physiologic) frequencies might be much softer.

In terms of clinical applications, having a method to assess the biological age of an individual patient's lens may allow a more fact-based decision of which type of refractive surgery would be most suitable for pre-presbyopic patients within an age range of 45 to 55 years. Furthermore, lens elastography opens a new way to advance lenticular refractive surgery (lentotomy[54]) and could permit an earlier detection of lenticular abnormalities[55] including cataracts.

In conclusion, the present study presents an approach to quantify the relative biomechanical stiffness of different lens regions and the GRIN distribution across the human crystalline lens in vivo and at the same time. This will allow a more detailed optomechanical assessment of the lens and might become a versatile tool for ophthalmologists and vision scientist to study the underlying principles of accommodation, the origin and progression of presbyopia, as well as ocular disorder such as cataract and high myopia.

## Data availability
The individual OCT data sets cannot be publicly disclosed to safeguard the privacy of the study participants, but are available from the corresponding author upon reasonable request and with permission from the Swiss Association of Research Ethics Committees. As part of this published article, Supplementary Data 1 contains source data supporting Figs. 2, 4, 5. The numerical data used to plot Fig. 2 a-d and 4 a-h can be found in Supplementary Data 2.

## Code availability
Inquiries regarding access to the software enabling raw data export from the Anterion OCT device should be directed to the manufacturer (Heidelberg Engineering). The remaining post-processing operations performed on the raw OCT signal are available on Zenodo[56].

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

## Acknowledgements
This work received funding from the European Union's HORIZON 2020 research and innovation programme under grant agreement No 956720, and from the AMBIZIONE career grant PZ00P2_174113 from the Swiss National Science Foundation. The authors thank Dr. Iulen Cabeza Gil for initial discussions.

## Author contributions
Conceived the idea (S.K.), designed the experimental protocol (S.K., M.F., E.A.T.N., F.H.), conducted the measurements (M.F., M.E.A., V.T., L.K.), developed and applied the data processing approach (S.K.), conducted the simulation (S.K.), drafted the manuscript (S.K.), revised the manuscript (M.F., M.E.A., V.T., E.A.T.N., L.K., F.H.).

## Competing interests
The author S.K. declares the following competing interest: Funding (research support: equipment from Heidelberg Engineering), Patent application (METHOD AND APPARATUS FOR INVESTIGATING A SAMPLE, EP19193554.3). All other authors declare no competing interests.
