## [Peer Review File · Communications Medicine]

referee reports: first round

Reviewers' comments:

Reviewer #1 (Remarks to the Author):

In this study, the authors present a novel methodology to quantify the gradient refractive index profile and deformation characteristics in the crystalline lens. I have enclosed below a few aspects that the authors should address:

1.The authors should explain the rationale behind including eyes with progressive keratoconus, and the value that this adds to the study.

Did subjects in the keratoconus group have corneal cross-linking? This would be relevant to know as this surgical treatment induces a corneal haze that could have affected the reflectometry.

How did the authors ensure that subjects with keratoconus had a proper accommodation response? (was the measurement taken with corrective lenses on?)

What is the basis for combining the results from the healthy and keratoconus groups? Would it be more relevant to present the results for each group individually? This point would be relevant to discuss in the Discussion.

2.The authors mentioned that 12 subjects were measured in this study. However, information about how many subjects were included in each age group is not available. Could the authors explain if this number has enough statistical power to extract general conclusions?

3.The study participants wore goggles during the scan acquisition, which introduced extra refractive indices. Could the authors explain in more detail how this was considered in the analysis, and if this could have affected the numerical simulation? Could the authors provide information relative to the goggles (design, refractive index, etc.)?

In lines 473-476, it can be read that the OCT beam is not orthogonally to the crystalline lens due to previous optical elements. Did the authors consider the goggle effect in the angle reported?

4.The Shapiro-Wilk test was used to assess the normality of the data. However, it is not specified if the data was normally distributed. Also, information about the statistical tests used to compare samples is missing. For example, in the Results section, we can find information about the t-test used, but this was not explained in the Statistical Analysis section.

5. A clearer description of the micro-fluctuations of accommodation needs to be included in the Method section. At least, knowing about the target design and vergence would be relevant. Also, the analysis of the micro-fluctuations needs to be described in more detail. For example, the micro-fluctuation value is presented in line 338 (i.e., 0.30 Hz), unfortunately, it is not known how this value was calculated.

Regarding the image acquisition, it is not the same to get 128 B-scans of the same area of the lens, as 128 radial B-scans. Could the authors specify this detail and how this could have affected the results?

Since micro-fluctuations were assessed, the information about the acquisition time needs to be included in the Method section. This information is indeed found in the Discussion section, line 462.

6. In lines 328-330, the values from the accommodation and dis-accommodation are presented. Could the authors define the term disaccommodation? Were the measurements taken while the eye was disaccommodating or just when the accommodation was relaxed?

Reviewer #2 (Remarks to the Author):

This manuscript presents an interesting method to examine the mechanical characteristics of the human crystalline lens and the GRIN distribution in vivo using OCT. An external ambient pressure change and the naturally occurring steady-state accommodative microfluctuations were used as two distinct mechanical stimuli.

There are few questions regarding the methodology and clarifications are needed on the same to be able to interpret the findings.

1. It is mentioned that previously collected data sets from an in vivo elastography studies in healthy participants and progressive keratoconus patients were used in this study. In total measurements from 12 participants in 6 age groups were included in the study. However, in the reference mentioned (34), only one eye from a healthy participant was included. It is not clear on from where the other 11 participant measurements were retrieved.

2. How many participants were included in each age group? Was it 2 per group? If not, were there any age groups over/under represented?

3. Why are subjects with keratoconus included in this study? How many participants had keratoconus out of the 12 included? It is not discussed anything further about keratoconus other than the methods. Are there any limitations in the analysis due to keratoconus, if not this should be mentioned in the discussion?

4. What fixation target was used during OCT measurements? Is it the internal fixation target in the OCT instrument? How can the micro-fluctuations and micro-accommodation and micro-disaccommodation be measured acceptably with that fixation target?

5. The lens geometry was divided into cortex and nucleus based on the reflectivity profile of the OCT image. What criteria was used to determine the nuclear region?

6. A symmetrically distributed GRIN was assumed for the refractive index mapping from the dataset. With OCT measurements, the entire posterior lens is not always visible, and how can this affect the estimation of refractive index?

Reviewer #3 (Remarks to the Author):

In the manuscript, the authors have evaluated both an external ambient pressure change and the naturally occurring steady-state micro-fluctuations of accommodation as two distinct mechanical stimuli for their suitability to be used in the context of optical coherence elastography in the in vivo human crystalline lens. The authors have determined the GRIN from the same data set. I have some major concerns regarding the studies as follows:

Major comments:

1. The major concern with the presented work is that it is unclear how ambient pressure changes induced strain in the crystalline lens. Moreover, the biomechanical properties of the lens are weakly related to the fluctuations in IOP.
2. The study introduces an innovative approach, but it is important to evaluate/acknowledge the limitations of this technology, particularly in terms of its resolution, sensitivity, and accuracy in differentiating between cortical and nuclear regions of the lens.
3. In line 165-176, the authors have an assumption about the symmetrical distribution of GRIN. They assumed that the sum of the axial gradient of the refractive index within the lens equals zero across an A-line. Does it make any sense?
4. With only 12 participants, the sample size is quite small. This raises questions about the statistical power of the study and the generalizability of its findings. Additionally, the demographic diversity of the sample should be scrutinized.

Minor comments:

1. In Figure 2, I would suggest adding the title to the color bars. It is mentioned in the caption of the figure that Fig. 2B and 2C show the strain amplitude, but it needs to be added to the figures. In Figures 3 A and B, the unit for the y-axis should be corrected. I believe it is a typo, and the authors meant to put %. Figure 4B, the unit for the y-axis should be corrected.
2. The manuscript should be well-written with a conclusive statement and should explain discrepancies in their trends.
3. The introduction to this paper is very lengthy and refers to techniques that are not directly related to the research discussed in this paper. For example, there is no major significance/contribution to the introduction by mentioning Brillouin Microscopy.
4. The title is very ambiguous.

Reviewer #4 (Remarks to the Author):

In this manuscript the authors present a novel type of optical coherence elastography that can examine the mechanical characteristics of the human crystalline lens and the GRIN distribution in vivo. The authors demonstrate proof of principle results by extracting lens displacement and strain measurements from an age-mixed group of human subjects in response to an external and an intrinsic mechanical deformation stimulus. Interestingly, they utilize the deformations induced by the recently discovered internal micro-fluctuations in axial length to calculate regional differences in stiffness in the lens cortex and nucleus. These in vivo measures confirm the ex vivo measurements that had shown in young accommodating lenses the cortex is stiffer than the nucleus, but with increasing age and the onset of presbyopia the nucleus becomes stiffer than the cortex.

The methods used in this study are also able to extract the GRIN distribution. While the shape of the GRIN obtained by the authors for young lens is comparable to other studies, in older lenses the well characterized flattening/plateau of the GRIN is not visible. Also, the authors report an increase in the maximum refractive index seen in the nucleus with age. This is not consistent with the concept of the lens paradox that predicts the observed hyperopic shift in lens power with age is driven by a reduction in the refractive index of the lens nucleus. Can the authors explain this inconsistency?

As a lens researcher who is not a bioengineer, but who has collaborated extensively with them, I found it difficult to work out how many of the results were obtained and how they relate to the structure and function of the lens. For example, your figures go straight to a comparison of the age-related changes in the parameters you have measured. This makes all the figures very busy and difficult to understand. I would suggest using a “representative” lens of one age to show how the data was collected. Then showing the age-related trends for the different age groups. The authors should consider how they present their data as the figures are too small, the colour schemes could be improved, and details of the lens geometry (anterior, centre, posterior) should be included to give an indication of origination. Also, the figure legends need to be more comprehensive.

Figure 1: Include on the x axis which are the anterior and posterior poles. Panel C the scale and colours need to be altered to see the data better.

Figure 2: The images need to be zoomed out to better show the anterior and posterior surfaces. I would suggest the authors try making the images bigger and to halve them to bring the dis-accommodated and accommodated images together to facilitate comparisons. In this format you could have only one common colour table and x y axis.

Figure 3: Panel A is too small and complex. Not sure if and how the values in panel B give rise to the ratio in panel C?

Figure 4: I am not sure what is being presented here?

Discussion: This section tends to jump around a bit and would benefit from a reorder. The limitations of the results need to be discussed, then their significance to our current understanding of lens accommodation and the development of presbyopia, and then finish with the future clinical application.

More effort needs to be made to discuss the findings in a language that can be understood by lens researchers by relating their findings back to the theories around the mechanisms driving accommodation and the subsequent development of presbyopia.

author responses: first round

Reviewer #1 (Remarks to the Author):

In this study, the authors present a novel methodology to quantify the gradient refractive index profile and deformation characteristics in the crystalline lens. I have enclosed below a few aspects that the authors should address:

1. The authors should explain the rationale behind including eyes with progressive keratoconus, and the value that this adds to the study.

Did subjects in the keratoconus group have corneal cross-linking? This would be relevant to know as this surgical treatment induces a corneal haze that could have affected the reflectometry.

How did the authors ensure that subjects with keratoconus had a proper accommodation response? (was the measurement taken with corrective lenses on?)

What is the basis for combining the results from the healthy and keratoconus groups? Would it be more relevant to present the results for each group individually? This point would be relevant to discuss in the Discussion.

R1. The reason for including keratoconus patients and healthy participants was that this measurement set has been recorded as part of another study and therefore was available to the authors. Per se, no differences in the crystalline lenses of these subgroups are expected, which is the reason why their results have been presented jointly. Also retrospectively, in those age categories in which a healthy and KC person have been measured, there were no evident differences between the two. These keratoconus patients have all been measured before and after receiving CXL treatment and the best measurement in terms of lens visibility was chosen. To verify the hypothesis of the reviewer that CXL could affect the reflectivity of the lens, we have assessed the reflectivity profile of the lens before and after the treatment and observed no consistent differences. Please note that even in the event that corneal haze would have an impact on the lens' reflectivity, it would lead to an overall decrease of the intensity. For the purpose of the current study, the relative (rather than absolute) reflectivity changes in the reflectivity profile were used as an indicator of the nuclear and cortical regions.

Measurements on keratoconus as on healthy individuals were taken using the internal fixation target of the OCT device. There was no further specific control or verification of the accommodative state, which is indeed a limitation of the study that is mentioned in the limitations. Nevertheless, we confidently may assume that the accommodative demand during the measurement was constant (which is confirmed by the minimal lens thickness changes of $\pm 3\mu\text{m}$ observed throughout the measurement duration).

The manuscript has been revised and clarified on the above-mentioned points.

2. The authors mentioned that 12 subjects were measured in this study. However, information about how many subjects were included in each age group is not available. Could the authors explain if this number has enough statistical power to extract general conclusions?

R2. Each of the six age groups was represented by two individuals. We conducted a post-hoc power calculation based on the observed correlations of cortical and nuclear strains (Fig 4G). With the sample size of 12, a power of 0.85 was achieved at an alpha error of 5%. A power of 0.8 or higher is generally recommended for statistical analyses. To compare difference in refractive index within different regions of the lens, a related samples Wilcoxon signed rank test was applied. The achieved statistical power of this analysis was 100%.

3. The study participants wore goggles during the scan acquisition, which introduced extra refractive indices. Could the authors explain in more detail how this was considered in the analysis, and if this could have affected the numerical simulation? Could the authors provide information relative to the goggles (design, refractive index, etc.)?

In lines 473-476, it can be read that the OCT beam is not orthogonally to the crystalline lens due to previous optical elements. Did the authors consider the goggle effect in the angle reported?

R3. The goggles were made of polycarbonate with a refractive index of 1.586 and a thickness of the front window of approx. 2mm. Inserting this layer in the OCT beam leads to a displacement of the eye towards the back by approx. $2\text{mm} \cdot (1.586 - 1) = 1.2\text{mm}$. The person consequently needs to place their head 1.2mm closer to the OCT objective lens than in a standard measurement. The amount of chromatic dispersion caused by the glasses was neglected, but may have led to a slight deterioration of the OCT's axial resolution. The important

thing to note here is that we determined the differences before and after applying a pressure difference or before and after undergoing the micro-fluctuation. Thus, the reference scan always included the goggle layer and therefore the latter should have not biased the measurements in any way. Given that the goggle front window was flat, no geometrical distortion was induced. Therefore, there is no reason to expect an effect on the numerical simulations. The manuscript has been revised and the requested information been added.

4. The Shapiro-Wilk test was used to assess the normality of the data. However, it is not specified if the data was normally distributed. Also, information about the statistical tests used to compare samples is missing. For example, in the Results section, we can find information about the t-test used, but this was not explained in the Statistical Analysis section.

R4. Thank you. This information has been included in the revised manuscript. In fact, the mentioning of a t-test was wrong – a related samples Wilcoxon signed rank test was performed to assess the respective differences. The manuscript has been corrected accordingly.

5. A clearer description of the micro-fluctuations of accommodation needs to be included in the Method section. At least, knowing about the target design and vergence would be relevant. Also, the analysis of the micro-fluctuations needs to be described in more detail. For example, the micro-fluctuation value is presented in line 338 (i.e., 0.30 Hz), unfortunately, it is not known how this value was calculated. Regarding the image acquisition, it is not the same to get 128 B-scans of the same area of the lens, as 128 radial B-scans. Could the authors specify this detail and how this could have affected the results? Since micro-fluctuations were assessed, the information about the acquisition time needs to be included in the Method section. This information is indeed found in the Discussion section, line 462.

R5. As recommended, we now provide a more detailed description of the target design and expected vergence in the revised manuscript. In addition, new Fig.1 provides an overview of the different processing steps, including the extraction of the frequency of the microfluctuations.

We fully agree that it is a big difference whether repeated B-scans are taken at the same position or radially distributed. The latter would not permit to maintain the phase reference necessary to conduct the present analysis. We now include the acquisition frequency in the methods section.

6. In lines 328-330, the values from the accommodation and dis-accommodation are presented. Could the authors define the term disaccommodation? Were the measurements taken while the eye was disaccommodating or just when the accommodation was relaxed?

R6. We now include more details about the status of the eye during the measurement (far vision, zero vergence). Micro-accommodation and micro-disaccommodation refer to oscillations around this value in the direction of accommodation and disaccommodation, respectively. Notice that there was no macroscopic change in the accommodative state during the whole measurement (assess via thickness changes). The following definition has been added in the results section:

In this context, micro-accommodation was defined as the lens becoming thicker and micro-disaccommodation as the lens becoming thinner. It is worth mentioning that lens micro-fluctuations were observed even at no accommodative demand.

Reviewer #2 (Remarks to the Author):

This manuscript presents an interesting method to examine the mechanical characteristics of the human crystalline lens and the GRIN distribution in vivo using OCT. An external ambient pressure change and the naturally occurring steady-state accommodative microfluctuations were used as two distinct mechanical stimuli.

There are few questions regarding the methodology and clarifications are needed on the same to be able to interpret the findings.

1. It is mentioned that previously collected data sets from an in vivo elastography studies in healthy participants and progressive keratoconus patients were used in this study. In total measurements from 12

participants in 6 age groups were included in the study. However, in the reference mentioned (34), only one eye from a healthy participant was included. It is not clear on from where the other 11 participant measurements were retrieved.

R1. We apologize for this confusion. The study on the remaining data set has been reported at the ESCRS conference 2023 (see abstract below). The corresponding article describing these findings is currently in preparation. We now provide the reference to this conference contribution in the revised manuscript.

ESCRS 2023 - Free paper

CORNEA

Topic: *Keratoconus and secondary ectasia*

Abstract Submission Identifier: ESCRS23-FP-3546

High Resolution Optical Coherence Elastography: Clinical Evaluation Of Normal And Keratoconus Corneas

M. Hillen¹, M. Frigelli², E. A. Torres-Netto^{1, 3}, M. E. Aydemir⁴, N. Lu^{4, 5}, F. Hafezi^{3, 4}, S. Kling^{2, 6}

¹Ophthalmology, ELZA Institute, Dietikon/Zurich, ²ARTORG Center for Biomedical Engineering Research, University of Bern, Bern, ³Ocular Cell Biology Laboratory, University of Zurich, ⁴Ophthalmology, ELZA Institute, Zurich, Switzerland, ⁵Faculty of Medicine and Health Sciences, University of Antwerp, Wilrijk, Belgium, ⁶OPTIC-team, Computer Vision Laboratory, ETH Zurich, Zurich, Switzerland

Do you want to apply for a Trainee Bursary?: No

I confirm that I am an Ophthalmologist: No

Purpose: Optical coherence elastography (OCE) is an emerging technology capable of detecting localized biomechanical alterations by inducing tissue deformation and concurrently capturing images through optical coherence tomography (OCT). In this study, we evaluated the potential of OCE, employing ambient pressure modulation for corneal deformation, to distinguish between normal and keratoconic corneas in vivo.

Setting: University of Zurich, CABMM; OPTIC team, Computer Vision Laboratory, ETH Zurich; ELZA Institute, Dietikon/Zurich, Switzerland

Methods: In this study, nine healthy participants and 15 patients with progressive keratoconus (KC) underwent OCE measurements. Customized swimming goggles, connected to an external pressure modulation unit, were worn by the participants. Over a period of approximately 2.6 seconds, 128 consecutive, repetitive B-scans were recorded. After 0.55 seconds, the goggle pressure decreased by 10 mmHg. The resulting corneal deformation was quantified using a phase-based displacement and strain computation methodology.

Results: The overall corneal strain exhibited positive values in KC cases and negative values in healthy corneas. By the end of the measurement, KC and healthy corneas had accumulated a posterior strain of $1.80 \pm 0.77\%$ and $-2.22 \pm 0.62\%$ ($p=0.001$), respectively. No significant difference was observed in anterior strain ($p=0.62$). In terms of the central cornea, anterior KC corneas displayed a tendency to move forward further on average compared to healthy corneas ($84 \pm 37\text{nm}$ vs. $-55 \pm 58\text{nm}$, $p=0.054$).

Conclusions: Optical coherence elastography demonstrates the ability to clinically distinguish between normal and keratoconic corneas by analyzing in-depth corneal strain. This technology enables the identification of localized biomechanical alterations in the cornea, and potentially paves the way for advances in keratoconus diagnosis and monitoring ectasia progression.

2. How many participants were included in each age group? Was it 2 per group? If not, were there any age groups over/under represented?

R2. Yes, two participants per age group were included.

3. Why are subjects with keratoconus included in this study? How many participants had keratoconus out of the 12 included? It is not discussed anything further about keratoconus other than the methods. Are there any limitations in the analysis due to keratoconus, if not this should be mentioned in the discussion?

R3. Five out of the 12 participants had keratoconus. Per se, keratoconus does not affect the crystalline lens and thus keratoconus and healthy participants should be considered equivalent for the purpose of this study. We have added a point in the methods and discussion addressing this matter.

4. What fixation target was used during OCT measurements? Is it the internal fixation target in the OCT instrument? How can the micro-fluctuations and micro-accommodation and micro-dis-accommodation be measured acceptably with that fixation target?

R4. Yes, the internal fixation light of the OCT system was used. In fact, the micro-accommodation and -disaccommodation were not measured with the fixation target, but with the OCT. These fluctuations occurred while a static fixation target was provided. The corresponding paragraph in the manuscript has been revised for clarity.

5. The lens geometry was divided into cortex and nucleus based on the reflectivity profile of the OCT image. What criteria was used to determine the nuclear region?

R5. When looking at the reflectivity profile of some individual lenses, the nuclear region was easier to identify, namely as the first and last valley in the reflectivity profile when moving from the anterior towards the posterior surface. In other lenses, the regions were more difficult to identify. Similarly, in the averaged reflectivity profile shown in Fig 1A, the anterior valley (situated at around -1.0) appears smeared out due to the averaging across several lenses. The posterior valley is better visible, but also less evident than in a single reflectivity profile. We have added the definition of the nuclear and cortical regions as a limitation in the discussion now.

6. A symmetrically distributed GRIN was assumed for the refractive index mapping from the dataset. With OCT measurements, the entire posterior lens is not always visible, and how can this affect the estimation of refractive index?

R6. Specifically, OCT measurements were selected for this study, in which at least 95% of the lens thickness were captured (refer to methods – data set, line 121). Therefore, the effect of the assumption of a symmetric distribution should not have notably affected the refractive index estimation.

Reviewer #3 (Remarks to the Author):

In the manuscript, the authors have evaluated both an external ambient pressure change and the naturally occurring steady-state micro-fluctuations of accommodation as two distinct mechanical stimuli for their suitability to be used in the context of optical coherence elastography in the in vivo human crystalline lens. The authors have determined the GRIN from the same data set. I have some major concerns regarding the studies as follows:

Major comments:

1. The major concern with the presented work is that it is unclear how ambient pressure changes induced strain in the crystalline lens. Moreover, the biomechanical properties of the lens are weakly related to the fluctuations in IOP.

R1. It is our understanding that the change in IOP provoked by the measurement induces a minimal change in the geometry of the lens (volume expansion due to a lower compressive force), but also of the ocular shell, which in turn results in a minimal change in the lens position with respect to the retina, inducing the need of a small degree of accommodation to keep the visual target in focus. We include these details now in the discussion. It is important to notice that we measure temporary deformations induced by a change to the environmental conditions and the lens needs to find its new equilibrium. Only due to the high (sub-pixel) sensitivity of the OCT to displacement and deformation it was possible to detect these minimal changes. In fact, we do not claim the biomechanical properties of the lens to be related to the IOP, but rather that IOP fluctuations result in small lenticular deformations (in the end the extent of these lenticular deformations depends on the biomechanical properties).

We have conducted a FEM analysis, which confirms the validity of the nature and magnitude of the lens deformations due to a change in the ambient pressure (and consequently IOP). We have added more details in the paragraphs describing the results and interpretation of the simulation.

2. The study introduces an innovative approach, but it is important to evaluate/acknowledge the limitations of

this technology, particularly in terms of its resolution, sensitivity, and accuracy in differentiating between cortical and nuclear regions of the lens.

R2. Thank you for raising this important point. We now mention the resolution of the images and limitations related to phase sensitive detection *in vivo*. We also discuss that the identification of cortical and nuclear regions is a limitation.

[...] were conducted with a spectral-domain commercial anterior segment optical coherence tomography (OCT) device (ANTERION, Heidelberg Engineering, Germany) with an axial and lateral resolution of 9.5 μm (in air) and 30 μm in the structural image

[...] We used phase-processing windows of the size $u_z = 2$ and $u_x = 5$, resulting in a resolution of 28 μm x 104 μm for displacement tracking.

[...] We used phase-processing windows of the size $w_z = 7$ and $w_x = 7$, resulting in a resolution of 97 μm x 146 μm in the cross-sectional displacement maps.

[...] $v_z = 15$ and $v_x = 15$ were the applied strain processing windows, resulting in a resolution of 207 μm x 313 μm in the cross-sectional strain maps.

Phase-sensitive detection itself poses a limitation with *in vivo* applications due to the high susceptibility to motion resulting in signal loss. Due to this effect, several frames in each acquisition were lost as shown in the visualization of $\Delta\varepsilon_z(t, z)$ in Fig. 1.

3. In line 165-176, the authors have an assumption about the symmetrical distribution of GRIN. They assumed that the sum of the axial gradient of the refractive index within the lens equals zero across an A-line. Does it make any sense?

R3. Yes, certainly. Having a look at previously derived GRIN distributions in the crystalline lens, a layered structure becomes evident with the lowest RI values in the anterior and posterior cortex and highest values in the center. Given that the same RI values are present at the anterior and posterior surfaces, the gradient of the RI (which is what the OCT measures) must be on average zero. A clarifying sentence has been added to the manuscript.

4. With only 12 participants, the sample size is quite small. This raises questions about the statistical power of the study and the generalizability of its findings. Additionally, the demographic diversity of the sample should be scrutinized.

R4. The achieved statistical power of the correlations was 85%, which is now mentioned in the manuscript. Also, we consider the demographic diversity is satisfactory in this data set, with a homogenous age stratification and a relatively similar ratio of healthy and keratoconus participants. We include these details now in the manuscript. Unfortunately, for data anonymization sex information was removed.

Minor comments:

1. In Figure 2, I would suggest adding the title to the color bars. It is mentioned in the caption of the figure that Fig. 2B and 2C show the strain amplitude, but it needs to be added to the figures. In Figures 3 A and B, the unit for the y-axis should be corrected. I believe it is a typo, and the authors meant to put %. Figure 4B, the unit for the y-axis should be corrected.

R1. Thank you. Figure 2 (now Fig.3) has been revised as recommended. The units in the y-axis of Figures 3A and B (now Fig.4) are correct. The observed strain is indeed very small. The same applies for Figure 4B (now Fig. 5).

2. The manuscript should be well-written with a conclusive statement and should explain discrepancies in their trends.

R2. Thank you. We have revised the discussion and conclusion accordingly.

3. The introduction to this paper is very lengthy and refers to techniques that are not directly related to the research discussed in this paper. For example, there is no major significance/contribution to the introduction by mentioning Brillouin Microscopy.

R3. We partially agree with the reviewer and have shortened the Brillouin section. We do believe that Brillouin microscopy deserves mentioning as a prior art of assessing the lens mechanics both, in vivo and with high resolution, in particular as we also refer to it in the discussion.

4. The title is very ambiguous.

R4. We have replaced the word *secret* by *mechanism* to make it less ambiguous.

Reviewer #4 (Remarks to the Author):

In this manuscript the authors present a novel type of optical coherence elastography that can examine the mechanical characteristics of the human crystalline lens and the GRIN distribution in vivo. The authors demonstrate proof of principle results by extracting lens displacement and strain measurements from an age-mixed group of human subjects in response to an external and an intrinsic mechanical deformation stimulus. Interestingly, they utilize the deformations induced by the recently discovered internal micro-fluctuations in axial length to calculate regional differences in stiffness in the lens cortex and nucleus. These in vivo measures confirm the ex vivo measurements that had shown in young accommodating lenses the cortex is stiffer than the nucleus, but with increasing age and the onset of presbyopia the nucleus becomes stiffer than the cortex.

The methods used in this study are also able to extract the GRIN distribution. While the shape of the GRIN obtained by the authors for young lens is comparable to other studies, in older lenses the well characterized flattening/plateau of the GRIN is not visible. Also, the authors report an increase in the maximum refractive index seen in the nucleus with age. This is not consistent with the concept of the lens paradox that predicts the observed hyperopic shift in lens power with age is driven by a reduction in the refractive index of the lens nucleus. Can the authors explain this inconsistency?

Thank you for raising this important point. According to our findings, with increasing age there was a trend towards higher refractive indices, particularly in the posterior nucleus. This is in agreement with previous literature describing the GRIN in older lenses to present a plateau in the central region. The lens paradox relates to the fact that the aging lens becomes less powerful despite its curvature becoming more curved. Two hypotheses to explain this behavior suggest either a reduced delta of the refractive index between cortex and periphery, or a steeper gradient refractive index in the periphery and a flatter gradient in the center. Our current observations are contrary to the former, but agree with the peripheral steepening of the gradient index of the latter. Nevertheless, the reviewer is right that the tendentially higher refractive index in the nucleus in old lenses contradicts both hypotheses. It is important to note that the RI observations at different age groups in the current study were merely a trend and not statistically significant. Therefore, we have decided to remove the comparison between younger and older individuals in this manuscript in order to not cause any confusion. This kind of analysis is likely better to be conducted with a larger sample size.

As a lens researcher who is not a bioengineer, but who has collaborated extensively with them, I found it difficult to work out how many of the results were obtained and how they relate to the structure and function of the lens. For example, your figures go straight to a comparison of the age-related changes in the parameters you have measured. This makes all the figures very busy and difficult to understand. I would suggest using a "representative" lens of one age to show how the data was collected. Then showing the age-related trends for the different age groups. The authors should consider how they present their data as the figures are too small, the colour schemes could be improved, and details of the lens geometry (anterior, centre, posterior) should be included to give an indication of origination. Also, the figure legends need to be more comprehensive.

Thank you for these valuable recommendations. We have added Fig.1, which provides an overview of the different post-processing steps and on the full set of measures that have been extracted. Furthermore, we have revised the remaining figures and legends.

Figure 1: Include on the x axis which are the anterior and posterior poles. Panel C the scale and colours need to be altered to see the data better.

The x-axis has been amended as recommended. Panel C was divided into panels C and D for better visualization.

Figure 2: The images need to be zoomed out to better show the anterior and posterior surfaces. I would suggest the authors try making the images bigger and to halve them to bring the dis-accommodated and accommodated images together to facilitate comparisons. In this format you could have only one common colour table and x y axis.

We have re-designed the figure with the reviewer's recommendations in mind.

Figure 3: Panel A is too small and complex. Not sure if and now the values in panel B give rise to the ratio in panel C?

Panel A has now been split up into six panels. Yes, the strains plotted in panel G give rise to the ratio plotted in panel H. For this, the average cortical strain is divided by the average nuclear strain at each age group. We now cite Fig.4H in the paragraph where the ratio is defined in order to make its relation to the figure clearer.

Figure 4: I am not sure what is being presented here?

This figure (now Figure 5) presents the results of the numerical simulation that replicate the experiments. Fig 5A is comparable to Fig 2C, and Fig 5B is comparable to Fig 4A-F. This relation is now also mentioned in the figure legend.

Discussion: This section tends to jump around a bit and would benefit from a reorder. The limitations of the results need to be discussed, then their significance to our current understanding of lens accommodation and the development of presbyopia, and then finish with the future clinical application.

More effort needs to be made to discuss the findings in a language that can be understood by lens researchers by relating their findings back to the theories around the mechanisms driving accommodation and the subsequent development of presbyopia.

Thank you. We have implemented the recommended reordering of the discussion. We have made some effort to relate to prior theories on accommodation and presbyopia.

referee reports: second round

REVIEWERS' COMMENTS:

Reviewer #1 (Remarks to the Author):

All my comments have been addressed

Reviewer #2 (Remarks to the Author):

All my comments have been addressed.

Reviewer #3 (Remarks to the Author):

The authors have addressed all comments and made constructive changes to increase the quality of the manuscript. The paper can be accepted in its present form.

Reviewer #4 (Remarks to the Author):

The authors have addressed my comments and improved the readability of this manuscript.

I did note the following:

Figure 1 is a good addition to the paper but it does not appear to be referred to in the relevant sections of the method

This passage "The gray shaded area in Fig. 1A indicates the nucleus. The remaining peripheral area was assigned as cortex." I believe this should now refer to Fig 2A

author responses: second round

REVIEWERS' COMMENTS:

Reviewer #1 (Remarks to the Author):

All my comments have been addressed

Reviewer #2 (Remarks to the Author):

All my comments have been addressed.

Reviewer #3 (Remarks to the Author):

The authors have addressed all comments and made constructive changes to increase the quality of the manuscript. The paper can be accepted in its present form.

Reviewer #4 (Remarks to the Author):

The authors have addressed my comments and improved the readability of this manuscript.

I did note the following:

Figure 1 is a good addition to the paper but it does not appear to be referred to in the relevant sections of the method

This passage "The gray shaded area in Fig. 1A indicates the nucleus. The remaining peripheral area was assigned as cortex." I believe this should now refer to Fig 2A

Thank you. The reviewer is right. We have revised the manuscript accordingly.